# Investigation and Characterization of Factors Affecting Rheological Properties of Poloxamer-Based Thermo-Sensitive Hydrogel

**DOI:** 10.3390/polym14245353

**Published:** 2022-12-07

**Authors:** I-Cheng Chen, Chen-Ying Su, Pei-Yu Chen, The Chien Hoang, Yi-Syue Tsou, Hsu-Wei Fang

**Affiliations:** 1Accelerator for Happiness and Health Industry, National Taipei University of Technology, No. 1, Sec. 3, Zhongxiao E. Rd., Taipei 10608, Taiwan; 2Department of Chemical Engineering and Biotechnology, National Taipei University of Technology, No. 1, Sec. 3, Zhongxiao E. Rd., Taipei 10608, Taiwan; 3Biotegy Vietnam Company Limited, No. 23, Alley 48, Tho Lao Street, Dong Mac Ward, Hai Ba Trung District, Hanoi City 11609, Vietnam; 4Department of Neurosurgery, Taipei Medical University Hospital, Taipei 110301, Taiwan; 5Taipei Neuroscience Institute, Taipei Medical University, Taipei 110301, Taiwan; 6The Ph.D. Program for Neural Regenerative Medicine, College of Medical Science and Technology, Taipei Medical University and National Health Research Institutes, Taipei 110301, Taiwan; 7Institute of Biomedical Engineering and Nanomedicine, National Health Research Institutes, No. 35, Keyan Road, Zhunan Town, Miaoli County 35053, Taiwan

**Keywords:** thermo-sensitive hydrogel, poloxamers, binary poloxamers, polysaccharides, sol-gel transition, biomaterials

## Abstract

Poloxamers are negatively temperature-sensitive hydrogels and their hydrophilic groups interact with water molecules at lower temperatures (liquid phase) while their hydrophobic groups interact more strongly with increases in temperature causing gelation. To investigate the factors affecting the rheological properties of poloxamers, various parameters including different poloxamer P407 concentrations, poloxamers P407/P188 blending ratios and additives were examined. The results presented a clear trend of decreasing gelling temperature/time when P407 was at higher concentrations. Moreover, the addition of P188 enhanced the gelling temperature regardless of poloxamer concentration. Polysaccharides and their derivatives have been widely used as components of hydrogel and we found that alginic acid (AA) or carboxymethyl cellulose (CMC) reduced the gelling temperature of poloxamers. In addition, AA-containing poloxamer promoted cell proliferation and both AA -and CMC-containing poloxamer hydrogels reduced cell migration. This study investigated the intriguing characteristics of poloxamer-based hydrogel, providing useful information to compounding an ideal and desired thermo-sensitive hydrogel for further potential clinical applications such as development of sprayable anti-adhesive barrier, wound-healing dressings or injectable drug-delivery system for cartilage repair.

## 1. Introduction

Hydrogels are hydrophilic polymers with three-dimensional (3D) porous structures that form cross-linked networks filtered with a large volume of water/biological fluid. Hydrogel technologies have a broad range of applications especially in biomedical fields such as drug-controlled delivery, wound dressings, biosensors, and tissue engineering due to their biodegradability, biocompatibility, low immunogenicity and ease of usage [1,2,3,4,5,6,7,8,9,10,11,12,13]. Hydrogels can be classified according to their origin, polymeric composition, configuration, type of cross-linking, physical appearance, and network electrical charge [14]. These hydrophilic polymer networks can be constructed by physical and/or chemical cross-linking (e.g., hydrogen bonding, metal coordination, hydrophobic interaction, redox-, thermal-, photo-, or radiation-initiated free radical polymerization) [15]. Among them, thermo-sensitive hydrogels have become the best-studied polymer systems for their controllable responses with external environmental changes.

According to their thermal responses, temperature-sensitive hydrogels can be divided into positively and negatively thermo-sensitive hydrogels [16,17]. Positively thermo-sensitive hydrogels (such as gelatin, agarose, amylose, amylopectin and some cellulose derivatives) have an upper critical solution temperature (UCST) and form gel upon cooling below this temperature [18,19]. By contrast, polymers with lower critical solution temperature (LCST) form negatively temperature-sensitive hydrogels as the temperature increases. For negatively temperature-sensitive hydrogels, hydrogen bonding between hydrophilic groups in the polymer with water molecules dominates to cause liquid phase at lower temperatures while hydrophobic interactions in the polymer become strengthened with the increasing temperature to induce a change in the solubility of the cross-linked network, causing gelation [20]. Currently, there are four categories of negatively temperature-sensitive hydrogels including synthetic poloxamer-based system, N-Isopropylacrylamide-based system (pNiPAAm), polyethylene glycol/poly(lactic-co-glycolic acid) (PEG/PLGA)-based system, and natural polysaccharides.

Poloxamers are a group of synthetic polymers possessing thermo-reversible properties that can be manipulated by adjusting polymer composition, molecular weight and concentration at physiological temperatures. Poloxamer-based polymers consist of hydrophilic ethylene oxide (EO) and hydrophobic propylene oxide (PO) blocks arranged in a triblock structure (PEO-PPO-PEO) and undergo sol-gel transition as temperature increases. Poloxamers contain different copolymers with a different number of hydrophilic EO and hydrophobic PO units, which are also characterized by their distinct hydrophilic–lipophilic balance (HLB) value [20]. Poloxamers have attracted notable attention for their enormous potential in various biomedical applications. Among various poloxamers, poloxamer 407 (P407, also called pluronic F127, PF127) has been commonly used due to its temperature sensitivity, biodegradable property and biocompatibility [21,22,23]. The average molecular weight of P407 is about 12,600 Da (9840–14,600 Da) [24] and its PEO content is 70%. P407 has been approved by the Food and Drug Administration (FDA) in licensed medicines and P407-based hydrogel has been considered safe and developed in wide range of applications such as drug delivery carrier, tissue regeneration scaffolds or micellar systems for gene delivery [20]. Nevertheless, P407 presents some limitations for certain applications as its insufficient gel strength and rapid dissolution were the significant drawbacks of this material [25].

Effects of additives on phase transitions of P407 have been reported in few studies [26]. Dumortier et al. reviewed some agents such as Na_2_CO_3_, PEG400, short-chain fatty acids and propanediol 1,2 which significantly modified the sol-gel transition temperature of P407 by rheological method [27]. Another study conducted by Seo et al. demonstrated the gelling temperature of P407/P188 (16%/10%) increased with increasing concentrations of surfactants (e.g., Sodium dodecyl sulfate, tween 20), cyclodextrin, alcohols (ethanol and propylene glycol) and 2% of MgCl_2_. By contrast, addition of dimethyl sulfoxide and NaCl lowered the gelling temperature [28]. In addition, different ratios of two poloxamers P407 and P188 (also called F68, molecular weight: 7680–9510 Da) were investigated to formulate in situ gelling systems for ocular drug delivery, and the results showed that these binary mixtures formed gels only above the physiological temperature while 20 wt % of P407 improved drug residence and was more suitable for ocular administration [29].

P407 has been widely investigated for various applications; however, only few studies have focused on systematic comparisons of different parameters and their effects on the thermo-gelation properties of P407-based hydrogel. To better understand how the gelling behavior of P407-based poloxamer hydrogel is influenced, poloxamer 188 and polysaccharides were blended and investigated. Poloxamers P407/P188 blending ratios and various concentrations of alginic acid (AA) and carboxymethyl cellulose (CMC) were examined. Cell proliferation and cell migration experiments performed with P407-based thermo-sensitive hydrogels offered useful information to develop ideal biomaterials for diverse pharmaceutical applications.

## 2. Materials and Methods

### 2.1. Hydrogel Components and Preparation

Briefly, to prepare thermo-sensitive hydrogel samples, required amount of poloxamers and additives were separately dissolved in 10 g of saline as a solvent (Taiwan Biotech, Taoyuan, Taiwan) and mixed by stirring at 4 °C for one day. All formulations are described in detail below in the Results section. Poloxamer 407 (Pluronic^®^ F-127, MW = 12,600 Da), AA (MW = 120,000–190,000 g/mol) and CMC (MW = 700,000 g/mol) were purchased from Sigma (St. Louis, MO, USA); poloxamer 188 (Kolliphor^®^ P-188, MW = 8400 Da) was purchased from BASF (Ludwigshafen am Rhein, Rheinland-Pfalz, Germany); hyaluronic acid (HA, MW = 7000 Da) was obtained from First Chemical (Taipei, Taiwan).

### 2.2. Rheology

The rheological properties of hydrogels were measured using a rheometer (MCR 302e, Anton Paar, Graz, Austria) at temperature from 10 °C to 40 °C at a heating speed of 0.05 °C per second. The measurement was performed at a 1 Hz frequency under 1% shear strain and a 1 mm gap. The storage modulus (G′) and loss modulus (G″) of each hydrogel were detected. The gelling temperature was defined as the temperature when solid gel state formed. The measurement was repeated three times for each experiment. Gelation time was determined by inverted tube method. A 2 mL volume of liquid sample was kept at 4 °C and then placed in a 37 °C water bath to obtain the time of the phase transition.

### 2.3. Viscosity Measurements

The viscosity of the various mixtures was measured using a programmable rheometer (MCR 302e, Anton Paar, Graz, Austria) with a programmable controller. The samples were rotated at 0.5 rpm over a temperature range of 25 to 37 °C. The measurement was repeated three times for each experiment.

### 2.4. Cell Culture

L929 cells (mouse fibroblasts, Strains number BCRC 60091) were purchased from Food Industry Research and Development Institute Taiwan (Hsinchu, Taiwan). Cells were routinely maintained in Modified Eagle Medium (MEM, Gibco, Thermo Fisher Scientific Inc., Waltham, MA, USA) supplemented with 10% fetal bovine serum (FBS, Invitrogen, Waltham, MA, USA) at 37 °C under 5% CO_2_ and 95% relative humidity.

### 2.5. Migration assays

To conduct cell migration study, ibidi Culture-Insert 2 Wells (ibidi GmbH, Grafelfing, Germany) were placed in 24-well plates and 70 μL of L929 cells were seeded with density of 2.8 × 10^4^ cells/well onto the wells of the inserts and cultured to 80% confluence. After cell attachment, both wells were filled with adherent cells, and the Culture-Insert was removed by using sterile tweezers to create a cell-free gap of 500 µm as the baseline value. The cells with gaps were then cultured with different hydrogels (poloxamers, poloxamers with AA or CMC) in Dulbecco’s Modified Eagle Medium (DMEM, Gibco, Thermo Fisher Scientific Inc., Waltham, MA, USA) at 37 °C under 5% CO_2_ and 95% relative humidity. The area of the gap was recorded every 24 h (0 to 72 h) and quantified by ImageJ software (National Institutes of Health) to evaluate cell migration rate. The measurement was repeated three times for each experiment.

### 2.6. Proliferation Assays

For the proliferation assay, L929 cells were seeded in the 96-well plate with 1 × 10^4^ cells/well density and co-cultured with 20 μL of hydrogels (poloxamers, poloxamers with AA or CMC) in 100 μL culture medium (DMEM supplemented with 10% FBS) for one and three days. CCK-8 reagent (Sigma, St. Louis, MO, USA) was added to the well on days 1 and 3 to evaluate the proliferation rate affected by different treatments. The samples were read by an Enzyme-linked immunosorbent assay (ELISA) reader (Tecan, Männedorf, Switzerland) with a wavelength of 460 nm to obtain OD values. Different numbers of L929 cells (0, 5000, 10,000, 15,000 and 20,000) were simultaneously plated in the 96-well plate and measured by CCK-8 reagent to create a standard curve for determination of corresponding cell numbers from OD 460 values. The measurement was repeated three times for each experiment.

### 2.7. Statistics

All data are represented as mean of three independent experiments. For statistical analysis, difference was evaluated by two-tailed Student’s *t*-test. A *p* value less than 0.05 was considered to be statistically significant.

## 3. Results

### 3.1. Characterization and Rheological Properties of the P407-Based Hydrogels

To investigate the thermo-sensitive sol-gel properties of the P407-based hydrogel, the first group of analyses examined the effects of P407 at different concentrations. Briefly, 2 to 4 g samples of P407 were prepared in 10 g of saline with 0.05 g of HA to increase viscosity and maintain mechanical integrity [30] (group P1 to P5 in Table 1). The sol-gel transition temperature was measured by the rheometer and defined as the temperature where G′ and G′′ were equal, representing same elastic and viscous properties while gelling temperature showed the stable gel state (Figure 1a). The P407 hydrogel was a liquid at 4 °C but quickly changed to gel state after being heated (Figure 1b). There was a clear trend of decreasing gelation temperature (ranging from 37.9 °C ± 0.35 °C to 13.8 °C ± 0.14 °C) when P407 was at higher concentrations (Figure 1c). Additionally, the phase transition time of the five concentrations of P407 (20% to 40% *w*/*w*, P1 to P5) was also examined, and the results demonstrated a negative correlation between P407 concentrations and gelation time. Figure 1d showed that only 30%, 35% and 40% (*w*/*w*) of P407 could form gel within 60 s and suitable for further investigation.

### 3.2. Effects of P188 on Rheological Properties of the P407-Based Hydrogels

According to gelling time obtained above, formulations P3, P4, P5 were selected for further investigation due to their proper gelling times. However, the gelling temperatures were too low (ranging from 19.1 °C to 13.8 °C, Figure 1c) for clinical usage. The most applied strategy to improve polymer characteristics is to combine additional polymers to gain new materials [31]. To this end, poloxamer 188 (P188, also known as pluronic F68) is a commonly used material that is also a nonionic block linear copolymer with rheological activities, possessing some cyto-protective functions in several tissue injury models [32].

To optimize the P407-based hydrogel system, P188 was blended with P407 at different ratios (wt %) with fixed total poloxamer concentration (P407/P188 = 9/1, 8/2, 7/3; Table 2), and the rheological properties were examined. As can be seen in Figure 2, there is a clear trend showing that P188 enhanced the gelling temperature regardless of total poloxamer concentrations. Moreover, 30~40% (*w*/*w*) of P188 only did not form gel below 37 °C (data not shown).

### 3.3. Effects of Polysaccharides on Rheological Properties of the P407/P188 Composite Hydrogel

Polysaccharides and their derivatives are natural polymers that can be obtained by fermentation and purification from animals, plants and microorganisms, and known to be widely investigated as additives cross-linked with hydrogels for tissue engineering [33,34,35,36,37,38,39]. Alginates are anionic natural polysaccharides derived from brown algae and bacteria and cellulose is neutral and the most abundant natural polymer extracted from cell wall of plants or produced by bacteria [40,41]. They are biocompatible polymers possessing gel-forming properties and mechanical stability, and often used for tissue regeneration, pharmaceutical excipients or drug delivery [42].

To explore the effects of polysaccharides as additives in poloxamer-based hydrogel, different amount of AA or CMC were mixed with P407 and P188 as shown in Table 3. Addition of AA or CMC (ranging from 1% to 3% *w*/*w*) reduced the gelling temperature of 30% (*w*/*w*) P407/P188 (8/2) hydrogel and greatly enhanced the viscosity of the gels, indicating that AA or CMC could improve the mechanical property of P407/P188 binary hydrogel (Figure 3).

Moreover, addition of AA or CMC reduced the gelling temperature when P407 and P188 were blended at 8/2 ratio regardless of poloxamer concentrations (Table 4, Figure 4a). In general, CMC containing gels displayed higher viscosity than gels with AA (Figure 3b and Figure 4b). Interestingly, the gelling time were also examined and the results showed that only AP7 and CP7 (35% of P407/P188 at 8/2 ratio containing 1% of AA or CMC, Table 4, Figure 4c) could form gel within 60 s and these formulations were used for further cell experiments.

### 3.4. Effects of Poloxamer-Based Hydrogel on Cell Proliferation and Migration

To explore the potential applications of poloxamer-based hydrogels developed in this study, AP7 and CP7 from Table 4 (35% of P407/P188 at 8/2 ratio containing 1% of AA or CMC) were used for evaluating cell migration and cell proliferation. The culture-inserts were used to create the gaps for evaluating the migration ability of treated cells. As shown in Figure 5a,b, 35% of P407/P188 with 1% AA (AP7) significantly promoted more cell migration than 35% of P407/P188 with 1% CMC (CP7), though the migration rate was much lower than untreated cells. Interestingly, AP7 also notably promoted cell proliferation at day 3 with a 2-fold increase in cell number compared to 35% P407/P188 binary mixture without AA (PP5) (Figure 5c).

## 4. Discussion

The most important property for a desired biomaterial to possess is good biocompatibility. In order to be commercially successful, chemical compatibility between drug and gel, manufacturing robustness, drug-loading capacity and storage stability are also important [43]. Towards this end, poloxamer P407 with thermo-responsive property evidences a history of use in humans and was selected as a base for formulations of thermo-sensitive hydrogels in the current study [26]. We systematically compared different parameters affecting the thermo-transition of P407-based hydrogel and investigated their effects on cellular behaviors.

Several hydrophilic molecules such as HA, cellulose derivatives or chitosan 23 have been incorporated into poloxamer gels as auxiliary agents for controlled drug release and/or improvement of their mucoadhesive capability [44]. HA is a high molecular weight polysaccharide and it has been approved by US FDA for implantation. In general, HA contributes to the maintenance of mechanical integrity and serves as a lubricant with strong hydration ability to promote wound healing and tissue regeneration with hemostatic ability [30]. Accumulated evidence has shown that the addition of HA to poloxamer exerted positive effects in various aspects. Li et al. developed an HA/poloxamer (HA-POL) hydrogel and found that HA-POL hydrogel possessed better air permeability, prevented bacteria (*E. coli*) invasion, and promoted wound healing as a wound dressing [45]. HA combined with P407 also simulated the regeneration of cartilage and showed sustained release of drug (KGF-2) to improve knee osteoarthritis in rats [46]. It has been suggested that HA interferes with the interplay between poloxamer and water, resulting in favorable interaction of poloxamer molecules and larger micellar dimensions [47]. The addition of 1% (*w*/*w*) HA in 18% (*w*/*w*) P407 did not lead to significant changes in the rheological properties [44]. However, when the concentration of HA was increased and 5% HA was added into 5% of P407, the gelling temperature shifted from 30 °C to 37 °C compared to 5% P407 only. We also examined the gelling temperature of 35% P407/P188 (*w*/*w* %) hydrogel with 0.5% HA or without HA, and the results showed there was no significant difference between these groups (unpublished data).

The shorter gelling time may facilitate lowering the risk of dilution with body fluid and the possibility of drainage [27]. The results of gelling time from 30%, 35%, 40% P407 were below 60 s which was proper, considering the working time for physicians during operation (P3, P4, P5 in Table 1, Figure 1d). However, the gelling temperature of P3, P4, P5 was too low for physiological conditions. The phenomenon of thermal response undergoes a delicate dynamical balance between the hydrophobic and hydrophilic portions of the polymer monomer, therefore, the gelling time and gelling temperature can be manipulated by blending different poloxamers [1,17]. P407 and P188 are the most commonly used binary mixtures for their good water solubility, solution clearance and safety [20]. To optimize ophthalmic formulations which may prolong drug residence in the precorneal area, mixtures with 20% P407, 1 to 6% P188 and 2% cysteine were prepared and characterized by an adequate temperature of gelification and an important gel strength. The results indicated that sol-gel transition temperature increased with P188 concentration (1 *w*/*w* % to 6 *w*/*w* %) from 21.8 °C to 30.2 °C (determined via bath thermostat) [48]. By measuring with rheometer, the data also showed that increasing P188 (Pluronic F68) concentration (from 0:20, 2:28, 4:16, 6:14, 8:12, 10:10 and 20:0 for P188/P407 ratios) in a total 20 wt % of poloxamer mixtures resulted in an increase in gelation temperature from 28.1 °C to 74.7 °C, and the trend was consistent with the findings of the current study (Figure 2) [29]. The other study evaluating in situ gel formulations for a sustained ocular delivery of ketorolac tromethamine (KT) demonstrated that mixtures of P407/P188 in 23:10 (*w*/*v*%) and 23:15 (*w*/*v*%) ratios were a promising system for KT delivered across the cornea [49]. In addition to ocular administration, in a study for vaginal delivery, Tgel increased along with the increase of P188 content with fixed P407 concentration and P407 (20% *w*/*w*)/P188 (5% *w*/*w*) hydrogel has been investigated as a amphotericin B (AmB) release thermogel system to stabilize AmB nanosuspension [50]. A possible explanation for these results might be that P188 is considered more hydrophilic and its micellization temperature is higher [51]. Addition of P188 to hydrogel causes a decrease in the storage modulus and a gain in the loss modulus [52]. Apart from mixing with P188, P407 has been also blended with other poloxamers. A mixed-poloxamer micelle (P333 and P407) system was developed to deliver hydrophobic drugs talazoparib and buparlisib intravenously in a 4T1 murine breast cancer model [53].

Polysaccharides are inexpensive materials with good biocompatibility which have been widely applied in biomedical hydrogels and used in tissue engineering [54]. In the current study, we used polysaccharides AA and CMC as additives and explored their effects on tuning thermal properties of poloxamer hydrogels. Alginates are anionic biopolymers and ionic cross-linking is the main force to form hydrogel [40,55]. Furthermore, alginate is known to be an excellent hemostatic polymer and more widely used in the medical field in recent years [56]. The common gelling process and physical link of alginate occurs by exchange of Na^+^ from alginate acid with divalent cations. Although alginate is not intrinsically thermo-sensitive, semi-interpenetrating polymer network (semi-IPN) alginate hydrogels with thermo-sensitive property for drug delivery carriers were prepared via in situ copolymerization of NIPAAm with poly(ethylene glycol)-co-poly (epsilon-caprolactone) (PEG-co-PCL) in the presence of sodium alginate [57]. Cellulose is the most abundant natural polymer and its derivatives are usually water soluble [58]. Methylcellulose (MC) is thermo-responsive with high sol-gel transition temperature around 60–80 °C that it requires combination with other thermo-responsive polymers. P407 and CMC have been combined as a transdermal drug delivery system for treating atopic dermatitis (AD). The presence of CMCs was found to distinctly improve the porous structure of the P407 matrix and improved the physical properties of P407 hydrogel. The formation of pore channels from P407/CMCs hydrogel allowed a significant increase in drug permeability across the skin, therefore, exerting the potential to lessen the symptoms of AD [59,60]. The results obtained in this study showed that AA or CMC decreased the gelling temperature when mixed with P407/P188 binary mixtures (Figure 3a). Our results also showed that implementing AA or CMC enhanced viscosity of poloxamer hydrogel (Figure 3b). This improved mechanical property may be due to multi-interaction of poloxamers and polysaccharide molecules to reinforce the binding strength of polymer chains.

Most poloxamer-based hydrogel-related literature has only focused on discussing how drugs released from the hydrogel affect cells; however, little research was found that investigated the effects of poloxamer-based hydrogel on cells and its pharmacological impacts. In this study, we showed that P407/P188 binary mixture with addition of AA notably promoted cell proliferation (Figure 5c), which can be further used for clinical application such as wound healing and tissue repair. Interestingly, 22% of P407 was used as a temporary embolic agent for temporary vascular occlusion during percutaneous endovascular procedures [61]. P188 (F68) or P407 (F127) has been proposed to assist early attachment and enhance the growth rate of human gingival fibroblasts with very low concentrations for facilitating early postsurgical wound healing [62]. Arévalo-Silva et al. conducted a study for cartilage regeneration with three groups of implants: a control group using only the five non-biodegradable polymers as endoskeletal scaffolds; the five polymers enveloped by P407 only; and the implants coated with P407 seeded with chondrocytes. Only implants coated by P407 hydrogel plus cells generated healthy new cartilage [63]. Methylcellulose (MC)-based thermo-sensitive hydrogel system showed low protein adsorption and enhanced cell adhesion which may be used as a robust delivery vehicle to injured CNS tissue for neural cell transplantation strategies [64].

P407-based poloxamer has been extensively investigated for use as drug delivery systems, tissue regeneration scaffolds and micellar system for various administration routes including ophthalmic, orthopedic, nasal, vaginal, rectal, oral, transdermal, and parenteral [20]. As P407/P188 binary hydrogel containing AA developed in this study presented the ability to promote cell proliferation (Figure 5c) and HA and P407 have been reported to exert positive effects on articular cartilage [46], this formulation shows great potential to be applied as injectable porous carrier to entrap cells or therapeutic drugs/molecules (e.g., stem cells, exosomes, growth factors) to treat cartilage defects [65]. Drug-release experiments will be designed and performed in the near future. On the other hand, treatment of CMC containing P407/P188 binary hydrogel showed very little cell migration (Figure 5a) which would be an excellent candidate for developing sprayable anti-adhesion barrier for post-operative adhesion. Further animal tests need to be performed for functional tests.

## 5. Conclusions

This study set out to determine the parameters affecting the rheological properties of P407-based hydrogels. In conclusion, we provide useful information to modify poloxamer-based thermo-sensitive hydrogels as desired biomaterial for different purposes. The tunable property of poloxamer-based hydrogels makes them highly attractive, potentially highlighting an insight into the future development of sprayable or injectable thermo-sensitive hydrogels to recapitulate the microenvironments in tissues for clinical applications such as cartilage repair, wound healing and minimally invasive surgeries.

## Figures and Tables

**Figure 1 polymers-14-05353-f001:**
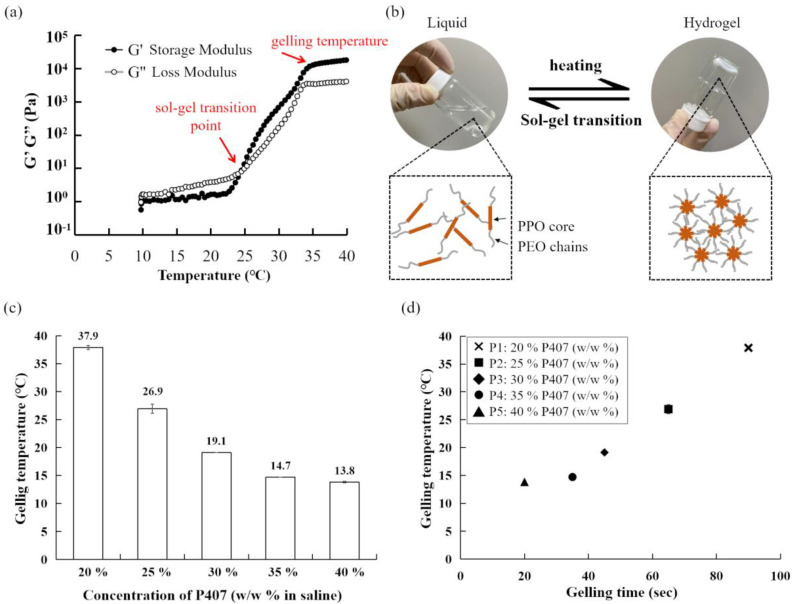
(**a**) Represented graphic for rheological behavior measured by the rheometer. Red arrow: the sol-gel transition temperature where G′ and G″ were equal, and gelling temperature; (**b**) sol-gel transition of P407-based hydrogel; (**c**) average gelling temperature of P407-based hydrogel at different concentrations of P407 (20, 25, 30, 35, 40% *w*/*w* of P407 in 10 g saline); (**d**) gelling time of P407-based hydrogel at different concentrations of P407.

**Figure 2 polymers-14-05353-f002:**
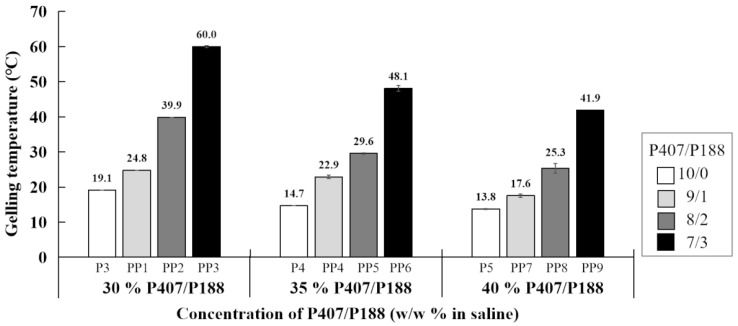
Average gelling temperature of P407/P188-blended hydrogel at different ratios. White bars: P407 only; gray bars: P407/P188 = 9/1; deep gray bars: P407/P188 = 8/2; black bars: P407/P188 = 7/3.

**Figure 3 polymers-14-05353-f003:**
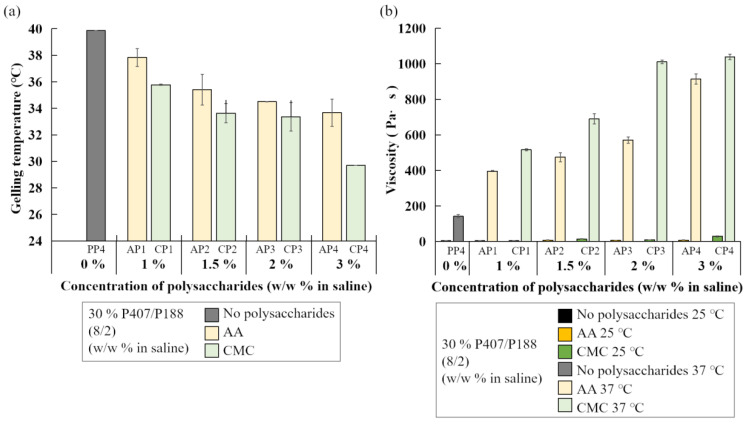
(**a**) Average gelling temperature and (**b**) average viscosity of P407/P188-blended hydrogels at 8/2 ratio without (gray bar) or with different concentrations of AA (yellow bars) or CMC (green bars).

**Figure 4 polymers-14-05353-f004:**
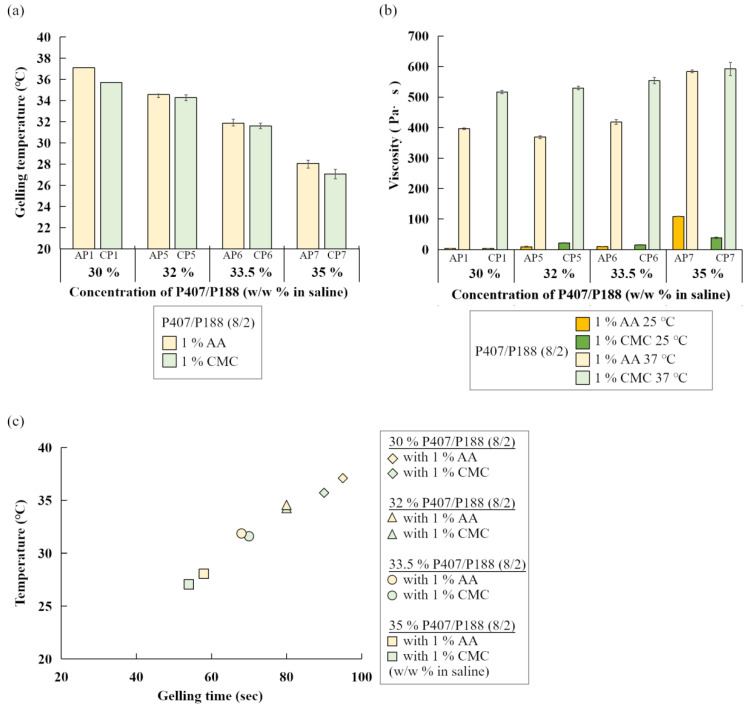
(**a**) Average gelling temperature, (**b**) average viscosity, and (**c**) gelation time of different concentrations of total poloxamers (30, 32, 33,5, 35% *w*/*w* of P407/P188 blended mixtures at 8/2 ratio) with 1% of AA (yellow bars, yellow marks) or CMC (green bars, green marks).

**Figure 5 polymers-14-05353-f005:**
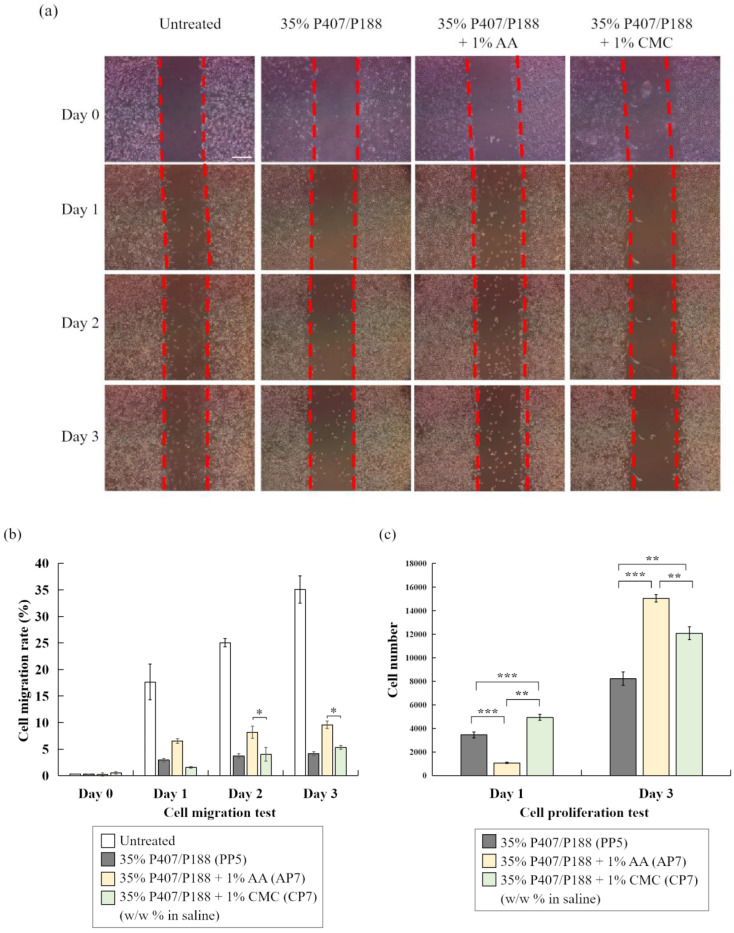
(**a**,**b**) Cell migration and (**c**) cell proliferation of L929 co-cultured with P407-based hydrogel. Gray bars: P407/P188 binary mixture (PP5); yellow bars: 35% P407/P188 with 1% AA (AP7); green bars: 35% P407/P188 with 1% CMC. Scale bar: 200 μm. * *p* < 0.05, ** *p* < 0.01 and *** *p* < 0.001.

**Table 1 polymers-14-05353-t001:** Components in the formulation of P407-based hydrogel (*w*/*w* % in 10 g saline).

Sample	Poloxamer 407 (*w*/*w* %)	HA ^1^ (*w*/*w* %)
P1	20	0.5
P2	25	0.5
P3	30	0.5
P4	35	0.5
P5	40	0.5

^1^ HA: Hyaluronic acid.

**Table 2 polymers-14-05353-t002:** Components in the formulation of P407-based hydrogel blended with different ratios of P188 (*w*/*w* % in 10 g saline).

Sample	P407/P188 (*w*/*w*)	P407 + P188 (*w*/*w* %)	HA (*w*/*w* %)
PP1	9/1	30	0.5
PP2	8/2	30	0.5
PP3	7/3	30	0.5
PP4	9/1	35	0.5
PP5	8/2	35	0.5
PP6	7/3	35	0.5
PP7	9/1	40	0.5
PP8	8/2	40	0.5
PP9	7/3	40	0.5

**Table 3 polymers-14-05353-t003:** Components in the formulation of P407-based hydrogel with different polysaccharides (*w*/*w* % in 10 g saline).

Sample	P407/P188 (*w*/*w*)	P407 + P188 (*w*/*w* %)	HA (*w*/*w* %)	AA (*w*/*w* %)	CMC (*w*/*w* %)
AP1	8/2	30	0.5	1.0	-
AP2	8/2	30	0.5	1.5	-
AP3	8/2	30	0.5	2.0	-
AP4	8/2	30	0.5	3.0	-
CP1	8/2	30	0.5	-	1.0
CP2	8/2	30	0.5	-	1.5
CP3	8/2	30	0.5	-	2.0
CP4	8/2	30	0.5	-	3.0

**Table 4 polymers-14-05353-t004:** Components in the formulation of P407-based hydrogel containing polysaccharides with different concentrations of poloxamers (*w*/*w* % in 10 g saline).

Sample	P407/P188 (*w*/*w*)	P407 + P188 (*w*/*w* %)	HA (*w*/*w* %)	AA (*w*/*w* %)	CMC (*w*/*w* %)
AP1	8/2	30.0	0.5	1	-
AP5	8/2	32.0	0.5	1	-
AP6	8/2	33.5	0.5	1	-
AP7	8/2	35.0	0.5	1	-
CP1	8/2	30.0	0.5	-	1
CP5	8/2	32.0	0.5	-	1
CP6	8/2	33.5	0.5	-	1
CP7	8/2	35.0	0.5	-	1

## Data Availability

The data presented in this study are available on request from the corresponding author.

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
