# Peer review of "Investigation and Characterization of Factors Affecting Rheological Properties of Poloxamer-Based Thermo-Sensitive Hydrogel"

_polymers, 2022, doi:10.3390/polym14245353_

Round 1

Reviewer 1 Report

This manuscript entitled “Investigation and characterization of factors affecting rheological properties of poloxamer-based thermos-sensitive hydrogel” investigated sol-gel transition influence factors of poloxamer-based negatively thermos-sensitive hydrogels. The authors have done a series of rheological and biological characterizations of polysaccharides containing poloxamer hydrogels. However, the structure of the article, the writing and the accuracy of the pictures are all flawed. Thus, this manuscript can be accepted after a major revision in the journal of Polymers. Furthermore, the following comments should be addressed before publication.

1. Please give more supporting literature to reinforce the point of “The hydrogel technologies have a broad range of applications, especially in biomedical fields such as drug controlled delivery, wound dressings, biosensors, and tissue engineering due to their biodegradability, biocompatibility, low immunogenicity and ease of usage.”

2. Please authors pay more attention to the manuscript writing, like “the average molecular weight of P407 is about 12,600” should be followed by the unit in the introduction part, and the “Insert wall” is misspelled as “Inert well” in part 3.4.

3. Please authors give out the molecular weight of AA, CMC, and HA.

4. Is the hydrogels used in migration assays including HA component?

5. It is recommended to use commercial hydrogel products as the control group in migration assays.

6. Please authors add the scale bar in Fig 5a.

7. As we all know; HA is a kind of polysaccharide. However, the first P407-based hydrogels are already started to use HA as thickeners. Please, authors, give more discussion to prove the reasonability of this basic hydrogel design and analysis of its effect on hydrogel rheological properties.

8. The concentration description of the hydrogels are X g / 10 mL saline, and it is quite difficult to compare with different experiment group. It is highly recommended to change the concentration description to W/V % or mg/mL.

9. Please authors give out the natural sol-gel transition temperature of poloxamer 188 in different polymer concentrations.

10. The cell proliferation test was assayed by the CCK-8 kit. However, the Y-axis of Fig 5c is cell number. Please authors give out the cell number calculation method by tested OD values.

11. It is strongly recommended that the authors further discuss the test results and rewrite the discussion section in comparison to the performance of the hydrogels in this work and similar hydrogels that have been published.

Author Response

Response to Reviewer 1 Comments

Thank you for the review of the manuscript entitled "Investigation and characterization of factors affecting rheological properties of poloxamer-based thermo-sensitive hydrogel" (polymers-2053528). We thank the constructive comments and have revised our manuscript accordingly in the text. The responses are highlighted in red font and listed below. Thank you for the prompt attention.

Yours sincerely,

Professor Hsu-Wei Fang, Ph.D. (Corresponding author)

Department of Chemical Engineering and Biotechnology, National Taipei University of Technology. Taipei 10608, Taiwan. (hwfang@ntut.edu.tw)

Reviewer 1 Comments

This manuscript entitled “Investigation and characterization of factors affecting rheological properties of poloxamer-based thermos-sensitive hydrogel” investigated sol-gel transition influence factors of poloxamer-based negatively thermos-sensitive hydrogels. The authors have done a series of rheological and biological characterizations of polysaccharides containing poloxamer hydrogels. However, the structure of the article, the writing and the accuracy of the pictures are all flawed. Thus, this manuscript can be accepted after a major revision in the journal of Polymers. Furthermore, the following comments should be addressed before publication.

Point 1: Please give more supporting literature to reinforce the point of “The hydrogel technologies have a broad range of applications, especially in biomedical fields such as drug controlled delivery, wound dressings, biosensors, and tissue engineering due to their biodegradability, biocompatibility, low immunogenicity and ease of usage.”

Response 1:

We have added more references for supporting the broad usages of hydrogel, please see the reference section:

  1. Deligkaris K., Tadele T.S., Olthuis W., van den Berg A. Hydrogel-based devices for biomedical applications. Sensors and Actuators B: Chemical. 2010;147(2):765-74.
  2. Sharma S., Tiwari S. A review on biomacromolecular hydrogel classification and its applications. International Journal of Biological Macromolecules. 2020;162:737-47.
  3. Tavakoli J., Tang Y. Hydrogel Based Sensors for Biomedical Applications: An Updated Review. Polymers. 2017;9(8):364.
  4. Tavakoli S., Klar A.S. Advanced Hydrogels as Wound Dressings. Biomolecules. 2020;10(8).
  5. Gong C.Y., Dong P.W., Shi S., Fu S.Z., Yang J.L., Guo G., et al. Thermosensitive PEG–PCL–PEG Hydrogel Controlled Drug Delivery System: Sol–Gel–Sol Transition and In Vitro Drug Release Study. Journal of pharmaceutical sciences. 2009;98(10):3707-17.
  6. Jeong B., Bae Y.H., Lee D.S., Kim S.W. Biodegradable block copolymers as injectable drug-delivery systems. Nature. 1997;388(6645):860-2.
  7. Lee J.W., Hua F.-j., Lee D.S. Thermoreversible gelation of biodegradable poly(ε-caprolactone) and poly(ethylene glycol) multiblock copolymers in aqueous solutions. Journal of Controlled Release. 2001;73(2):315-27.
  8. Liang X., Kozlovskaya V., Chen Y., Zavgorodnya O., Kharlampieva E. Thermosensitive Multilayer Hydrogels of Poly(N-vinylcaprolactam) as Nanothin Films and Shaped Capsules. Chemistry of Materials. 2012;24(19):3707-19.
  9. Vihola H., Laukkanen A., Tenhu H., Hirvonen J. Drug release characteristics of physically cross-linked thermosensitive poly(N-vinylcaprolactam) hydrogel particles. Journal of pharmaceutical sciences. 2008;97(11):4783-93.
  10. Zhang Q., Dong P., Chen L., Wang X., Lu S. Genipin-cross-linked thermosensitive silk sericin/poly(N-isopropylacrylamide) hydrogels for cell proliferation and rapid detachment. 2014;102(1):76-83.

Point 2: Please authors pay more attention to the manuscript writing, like “the average molecular weight of P407 is about 12,600” should be followed by the unit in the introduction part, and the “Insert wall” is misspelled as “Inert well” in part 3.4.

Response 2: 

Thanks for the reminder. We have added the unit Da for molecular weight. To avoid misunderstanding, we replaced “insert-well” to “culture-inserts”. Culture-insert is a device with two wells  to create a gap of 500 µm between cells. (section 3.4, line 253). We also corrected some small grammar mistakes in the article.

Point 3: Please authors give out the molecular weight of AA, CMC, and HA.

Response 3: 

The molecular weight of AA used in this study is about 120000-190000 g/mol, the average  molecular weight of CMC is 700000 g/mol and the average molecular weight of HA is approximately 7000 Da. We also added these information in the manuscript, please see section 2.1, line 114-117.

Point 4: Is the hydrogels used in migration assays including HA component?

Response 4: 

Yes, the hydrogel used in migration assays contained HA. Please see Table 2 PP5 (as control) and Table 4 AP7 and CP7.

Sample

P407/P188 (w/w)

P407 + P188 (w/w %)

HA (w/w %)

AA (w/w %)

CMC (w/w %)

PP5

8/2

35.0

0.5

-

-

AP7

8/2

35.0

0.5

1

-

CP7

8/2

35.0

0.5

-

1

Point 5: It is recommended to use commercial hydrogel products as the control group in migration assays.

Response 5: 

It is a great comment to compare commercial hydrogel with our formulations. However, so far no commercialized P407/P188 binary hydrogel products is available here in Taiwan. One commercialized barrier gel from FzioMed company contains P407 and CMC which is different from our design and it does not display thermo-sensitive property.  

Point 6: Please authors add the scale bar in Fig 5a.

Response 6: 

We have added the scale bar (200 μm) in Fig 5a. Please check Page 9, Fig. 5a.

Point 7: As we all know; HA is a kind of polysaccharide. However, the first P407-based hydrogels are already started to use HA as thickeners. Please, authors, give more discussion to prove the reasonability of this basic hydrogel design and analysis of its effect on hydrogel rheological properties.

Response 7: 

We have added a paragraph in discussion, please check line 274-294:

Several hydrophilic molecules such as HA, cellulose derivatives or chitosan 23 have been incorporated into poloxamer gels as auxiliary agents for controlled drug release and/or improvement of their mucoadhesive capability [44]. HA is a high molecular weight polysaccharide and it has been approved by US FDA for implantation. In general, HA contributes to maintaining the mechanical integrity and serves as lubricant with strong hydration ability to promote wound healing and tissue regeneration with hemostatic ability [30]. Accumulated evidences have shown that addition of HA in poloxamer exerted positive effects from various aspects. Li et al. developed a HA/poloxamer (HA-POL) hydrogel and found that HA-POL hydrogel possessed better air permeability, prevented bacteria (E. coli) invasion, and promoted wound healing as a wound dressing [45]. HA combining P407 also simulated the regeneration of cartilage and showed sustained release of drug (KGF-2) to improve knee osteoarthritis in rats [46]. It has been suggested that HA interferes interplay between Poloxamer and water, resulting in favorable interaction of poloxamer molecules and larger micellar dimensions [47]. The addition of 1% (w/w) HA in 18 % (w/w) P407 did not lead to significant changes in the rheological properties [44]. However, when the concentration of HA was increased and 5 % HA was added into 5 % of P407, the gelling temperature shifted from 30 C to 37 C compared to 5 % P407 only. We also examined the gelling temperature of 35 % P407/P188 (w/w%) hydrogel with 0.5 % HA or without HA, and the results showed there was no significant difference between these groups (unpublished data).   

Figure. Gelling temperature of P407/P188 hydrogels with or without HA (unpublished data).

Point 8: The concentration description of the hydrogels are X g / 10 mL saline, and it is quite difficult to compare with different experiment group. It is highly recommended to change the concentration description to W/V % or mg/mL.

Response 8: 

Thanks for the comment. To be clear, we replaced all the concentration description to w/w % in the text and figures.

Point 9:  Please authors give out the natural sol-gel transition temperature of poloxamer 188 in different polymer concentrations.

Response 9: 

We have tested the the gelling temperature of P188. the gelling temperature of 35 % (w/w) P188 was around 39.1 °C and 40 % P188 could not form gel. We added these observations in section 3.2, line 205-206. These results are similar to the description in other references:

  1. Ban, E.; Park, M.; Jeong, S.; Kwon, T.; Kim, E.-H.; Jung, K.; Kim, A. Poloxamer-Based Thermoreversible Gel for Topical Delivery of Emodin: Influence of P407 and P188 on Solubility of Emodin and Its Application in Cellular Activity Screening. Molecules 2017, 22, 246. https://doi.org/10.3390/molecules22020246
  2. Ji-Lai Tian, Ying-Zheng Zhao, Zhuo Jin, Cui-Tao Lu, Qin-Qin Tang, Qi Xiang, Chang-Zheng Sun, Lu Zhang, Yan-Yan Xu, Hui-Sheng Gao, Zhi-Cai Zhou, Xiao-Kun Li & Ying Zhang (2010) Synthesis and characterization of Poloxamer 188-grafted heparin copolymer, Drug Development and Industrial Pharmacy, 36:7, 832-838, DOI: 10.3109/03639040903520983

Point 10: . The cell proliferation test was assayed by the CCK-8 kit. However, the Y-axis of Fig 5c is cell number. Please authors give out the cell number calculation method by tested OD values.

Response 10: 

We have added the missing information in section 206, line 158-161: Different numbers of L929 cells (0, 5000, 10000, 15000 and 20000) were simultaneously plated in the 96-well plate and measured by CCK-8 reagent to create a standard curve for determination of corresponding cell numbers from OD 460 values.

Point 11: It is strongly recommended that the authors further discuss the test results and rewrite the discussion section in comparison to the performance of the hydrogels in this work and similar hydrogels that have been published.

Response 11: 

We have re-written the discussion section and added more comparisons regarding the properties of the hydrogels. Please see section 4.

Reviewer 2 Report

This is a very sound manuscript. In this work, the author systematically studied the effect HA, P188, AA and CMC on the rheological property of P407 based hydrogel. In addition, the author demonstrated those factor can be used to manipulate the cell proliferation and migration on produced P407 hydrogels. I would recommend the author to discussion more on the rationality to choose these factors (HA, P188, AA and CMC), which could significantly increase the scientific significant of this work. Otherwise, this is a great manuscript. 

Author Response

Response to Reviewer 2 Comments

Thank you for the review of the manuscript entitled "Investigation and characterization of factors affecting rheological properties of poloxamer-based thermo-sensitive hydrogel" (polymers-2053528). We thank the constructive comments and have revised our manuscript accordingly in the text. The responses are highlighted in red font and listed below. Thank you for the prompt attention.

Yours sincerely,

Professor Hsu-Wei Fang, Ph.D. (Corresponding author)

Department of Chemical Engineering and Biotechnology, National Taipei University of Technology. Taipei 10608, Taiwan. (hwfang@ntut.edu.tw)

Reviewer 2 Comments

This is a very sound manuscript. In this work, the author systematically studied the effect HA, P188, AA and CMC on the rheological property of P407 based hydrogel. In addition, the author demonstrated those factor can be used to manipulate the cell proliferation and migration on produced P407 hydrogels. I would recommend the author to discussion more on the rationality to choose these factors (HA, P188, AA and CMC), which could significantly increase the scientific significant of this work. Otherwise, this is a great manuscript.

Response :

Thanks for the comments. We have re-written the discussion section and added more discussion regarding the factors we selected for this study. Please see section 4.
